# Safety and Immunogenicity of the Monovalent Omicron XBB.1.5-Adapted BNT162b2 COVID-19 Vaccine in Individuals ≥12 Years Old: A Phase 2/3 Trial

**DOI:** 10.3390/vaccines12020118

**Published:** 2024-01-24

**Authors:** Juleen Gayed, Oyeniyi Diya, Francine S. Lowry, Xia Xu, Vishva Bangad, Federico Mensa, Jing Zou, Xuping Xie, Yanping Hu, Claire Lu, Mark Cutler, Todd Belanger, David Cooper, Kenneth Koury, Annaliesa S. Anderson, Özlem Türeci, Uǧur Şahin, Kena A. Swanson, Kayvon Modjarrad, Alejandra Gurtman, Nicholas Kitchin

**Affiliations:** 1Vaccine Research and Development, Pfizer Ltd., Hurley SL6 6RJ, UKnicholas.kitchin@pfizer.com (N.K.); 2Vaccine Research and Development, Pfizer Inc., Collegeville, PA 19426, USAxia.xu3@pfizer.com (X.X.); vishvaraj.bangad@pfizer.com (V.B.); 3BioNTech, 55131 Mainz, Germany; 4Department of Biochemistry & Molecular Biology, The University of Texas Medical Branch, Galveston, TX 77555, USAxuxie@utmb.edu (X.X.);; 5Vaccine Research and Development, Pfizer Inc., Pearl River, NY 10965, USAtodd.belanger@pfizer.com (T.B.); kayvon.modjarrad@pfizer.com (K.M.);

**Keywords:** BNT162b2, COVID-19, SARS-CoV-2, vaccine, variant-adapted, booster, Omicron, XBB.1.5

## Abstract

Vaccination remains an important mitigation tool against COVID-19. We report 1-month safety and preliminary immunogenicity data from a substudy of an ongoing, open-label, phase 2/3 study of monovalent Omicron XBB.1.5-adapted BNT162b2 (single 30-μg dose). Healthy participants ≥12 years old (*N* = 412 (12–17 years, *N* = 30; 18–55 years, *N* = 174; >55 years, *N* = 208)) who previously received ≥3 doses of a US-authorized mRNA vaccine, the most recent being an Omicron BA.4/BA.5-adapted bivalent vaccine ≥150 days before study vaccination, were vaccinated. Serum 50% neutralizing titers against Omicron XBB.1.5, EG.5.1, and BA.2.86 were measured 7 days and 1 month after vaccination in a subset of ≥18-year-olds (N = 40) who were positive for SARS-CoV-2 at baseline. Seven-day immunogenicity was also evaluated in a matched group who received bivalent BA.4/BA.5-adapted BNT162b2 in a previous study (ClinicalTrials.gov Identifier: NCT05472038). There were no new safety signals; local reactions and systemic events were mostly mild to moderate in severity, adverse events were infrequent, and none led to study withdrawal. The XBB.1.5-adapted BNT162b2 induced numerically higher titers against Omicron XBB.1.5, EG.5.1, and BA.2.86 than BA.4/BA.5-adapted BNT162b2 at 7 days and robust neutralizing responses to all three sublineages at 1 month. These data support a favorable benefit-risk profile of XBB.1.5-adapted BNT162b2 30 μg. ClinicalTrials.gov Identifier: NCT05997290

## 1. Introduction

Since late 2021, the SARS-CoV-2 epidemiologic landscape has been dominated by successive cycles of emergent Omicron sublineages [1], some with the ability to evade infection or vaccine-acquired host immunity [2]. In August 2022, Omicron sublineages XBB and XBB.1 were identified [3], and by early 2023, Omicron XBB.1.5 was predominant globally [4]. Omicron sublineages continue to emerge: in October 2023, EG.5.1 was predominant globally, and since the end of November 2023, BA.2.86 and JN.1 have been expanding rapidly [5,6,7]. Because vaccination remains critical to protect against the serious consequences of COVID-19 [8], vaccines will likely need to continue to adapt to this ongoing antigenic drift [2,9].

BNT162b2 is an mRNA-based COVID-19 vaccine that, as of December 2022, has been approved or authorized in more than 149 countries for individuals from 6 months of age [10,11,12]. The original version of BNT162b2 encodes the ancestral Wuhan-Hu-1 strain spike glycoprotein [13] and provided protection against COVID-19 caused by the SARS-CoV-2 ancestral strain and early variants, especially against severe disease [14,15,16]. The original vaccine was less protective against the Omicron variant [17,18], and Omicron-adapted vaccines, which contained mRNA components for the ancestral strain of the virus in combination with those encoding the Omicron BA.4/BA.5 spike protein, were subsequently widely introduced for the 2022 season [19]. Omicron BA.4/BA.5-adapted mRNA vaccines provide early effective protection against Omicron-related COVID-19, including against the XBB/XBB.1.5 sublineage [20,21]. Moreover, an additional BA.4/BA.5-adapted mRNA vaccine dose has been reported to have higher efficacy against severe Omicron-related illness than the original vaccine [22], indicating that better strain-matched vaccines improve protection against COVID-19. However, waning effectiveness of COVID-19 vaccines has been observed from 2 to 6 months after vaccination, including against severe disease [22,23,24].

In response to the continued evolution of SARS-CoV-2 Omicron sublineages, in May and June 2023, the World Health Organization and the US Food and Drug Administration (FDA), respectively, recommended that an XBB.1 monovalent variant vaccine be developed for the 2023/2024 season [25,26]. One approach suggested by the FDA was to use an XBB.1 descendent virus, such as XBB.1.5 [25]. Subsequently, in September 2023, the FDA authorized an updated monovalent BNT162b2 vaccine encoding the viral spike protein of SARS-CoV-2 Omicron XBB.1.5 (XBB.1.5-adapted BNT162b2) for use in individuals ≥6 months of age [11,27].

The aim of this analysis is to provide 1-month safety and preliminary immunogenicity data from a clinical trial investigating the safety and immunogenicity of the XBB.1.5-adapted BNT162b2. The trial consists of two substudies: one in vaccine-experienced individuals, and the other in previously unvaccinated individuals. Here, we present results in a population of vaccine-experienced individuals.

## 2. Materials and Methods

### 2.1. Study Design and Participants

In this ongoing, open-label, phase 2/3 study, participants received a single, 30 μg dose of XBB.1.5-adapted BNT162b2 (ClinicalTrials.gov Identifier: NCT05997290). It was planned that approximately 400 participants would be enrolled (200 participants from 12 to 55 years of age (including ≤50 participants from 12 to 17 years) and 200 participants >55 years of age). This report provides 1-month safety data and preliminary immunogenicity data from a substudy that included only vaccine-experienced individuals. The study enrolled healthy individuals who were at least 12 years old and who had previously received at least 3 doses of an mRNA COVID-19 vaccine that was authorized by the US Food and Drug Administration. In addition to receiving at least 3 doses of an mRNA COVID-19 vaccine, participants may have received other COVID-19 vaccine platforms. The most recent dose must have been the bivalent Omicron BA.4/BA.5-adapted vaccine, given at least 150 days before study vaccination. Immunocompromised individuals, those with a history of severe reaction associated with vaccination, and pregnant and breastfeeding individuals were excluded.

An institutional review board or independent ethics committee reviewed and approved relevant study documents, including the protocol. Study conduct adhered to the Declaration of Helsinki principles, the Council for International Organizations of Medical Sciences international ethical guidelines, and all applicable laws and regulations. Written informed consent was obtained from all participants; if a child or adolescent participant’s parent(s) or legal guardian(s) provided consent, the participant’s assent was also obtained if the participant was capable of providing assent. If study participants reached adulthood or the age of assent (per local institutional review board or ethics committee requirements) during the study, the child or adolescent then provided the appropriate consent or assent to document their willingness to continue in the study.

### 2.2. Objectives, Endpoints, and Assessments

Describing the tolerability and safety profile of XBB.1.5-adapted BNT162b2 at the 30-μg dose level in vaccine-experienced ≥12-year-old participants was the primary safety objective. Per protocol, the associated tolerability and safety endpoints included the proportions of participants reporting local reactions and systemic events up to 7 days after vaccination, adverse events through 1 month after vaccination, and serious adverse events through 6 months after vaccination. Described here are local reactions and systemic events up to 7 days following vaccination and adverse events and serious adverse events reported through 1 month after vaccination. Participants recorded local reaction and systemic event data using an electronic diary. If these local reactions or systemic events were not recorded in the electronic diary, they were reported as adverse events. Appendix A summarizes the grading scales used to describe the severity of local reactions and systemic events.

Protocol-specified adverse events of special interest included confirmed diagnoses of either myocarditis or pericarditis that occurred within 6 weeks after vaccination and potential menstrual cycle disturbances. Additional adverse events were reported as being of special interest as specified in a targeted medical event list that considered identified pharmacology and toxicology findings and possible class effects from sources such as the published literature and safety data assessment signals. A participant reporting any symptom(s) possibly indicative of myocarditis or pericarditis, such as acute chest pain, shortness of breath, or palpitations, within 6 weeks after receiving the study vaccination was to undergo cardiologist evaluation for diagnosis of potential myocarditis or pericarditis. Any diagnosis of myocarditis or pericarditis would be considered an important medical event and reported as a serious adverse event. Any study participant who reported any symptoms that may indicate a disturbance of their normal menstrual cycle (including, but not exclusively, heavy menstrual bleeding, amenorrhea, irregular periods) after receipt of study intervention until 6 months after vaccination would be specifically evaluated by the investigator. The symptoms, menstrual history, and any investigative results were recorded. Surveillance occurred throughout the study for any potential COVID-19 cases in all participants and multisystem inflammatory syndrome in children (MIS-C) in participants younger than 21 years. Described here are protocol-defined adverse events of special interest reported within 1 month after vaccination.

In this analysis, immunogenicity was evaluated in a subset (the variant neutralization subset) of 40 baseline SARS-CoV-2-positive participants (20 participants 18–55 years old and 20 participants >55 years old) who were selected at random. A qualified fluorescent focus reduction neutralization test (FFRNT) evaluated SARS-CoV-2 serum neutralization titers against Omicron XBB.1.5, EG.5.1, and BA.2.86 viruses. Immunogenicity endpoints included geometric mean titers (GMTs) 7 days after and 1 month after vaccination, geometric mean fold rises (GMFRs) from before to 7 days after and from before to 1 month after vaccination, and percentages of participants with seroresponses (defined as ≥4-fold rise from baseline or ≥4 × lower limit of quantitation (LLOQ) for baseline measurements <LLOQ) 7 days after and 1 month after vaccination. Seven-day immunogenicity was also evaluated in a group of 40 baseline SARS-CoV-2-positive participants who received bivalent BA.4/BA.5-adapted BNT162b2 in a previous study (ClinicalTrials.com Identifier: NCT05472038). The 7-day and 1-month samples were tested at different times; baseline samples were retested with the 1-month samples. Participants were matched by age, sex, and SARS-CoV-2 status to those in the current analysis. The inclusion criteria for the BA.4/BA.5-adapted cohort required the participant to have received 3 or 4 previous doses of original BNT162b2 30 μg with the last dose being 150 to 365 days before study vaccination.

### 2.3. Statistical Analysis

Descriptive statistics, which include the counts and percentages of participants, as well as the associated 95% CIs determined by the Clopper-Pearson method, are provided for each reactogenicity endpoint by age subgroup (12–17 years, 18–55 years, >55 years) and overall. Adverse events and serious adverse events were categorized using terminology of the *Medical Dictionary for Regulatory Activities* v26.0 and presented descriptively by age subgroup and overall. All participants receiving the study vaccine are included in the safety population.

The GMTs were calculated by exponentiating the mean and the GMFRs by exponentiating the mean of the difference of the logarithmically transformed assay results and corresponding Student *t* distribution-based 2-sided 95% CIs. The percentages of participants with a seroresponse are presented descriptively with associated Clopper-Pearson 95% CIs. The evaluable immunogenicity population was used for the 1-month analysis and comprised individuals from the variant neutralization subset who received the study vaccine, had at least one valid and determinate immunogenicity result using the blood sample that was collected within 28–42 days after vaccination, and who had no other clinician-determined important protocol deviations. The all-available immunogenicity population was used for the 7-day analysis and comprised participants who received the study vaccine and who had a valid and determinate postvaccination immunogenicity result. An evaluable immunogenicity population was not defined for the 7-day analysis.

## 3. Results

### 3.1. Participants

This substudy was conducted at 20 sites in the United States. From 10 August 2023, to data cutoff (27 September 2023), 412 participants received the XBB.1.5-adapted BNT162b2 30 μg (Figure 1; 12–17 years of age, *N* = 30; 18–55 years of age, *N* = 174; >55 years of age, *N* = 208).

Overall, 58.7% of participants were female, 79.1% were White, 12.6% were Black, and 18.2% were Hispanic or Latino (Table 1). Median age at vaccination was 14.0 years in the 12–17-year-old group, 42.0 years in the 18–55-year-old group, and 68.5 years in the >55-year-old group. Overall, 78.2% of participants were baseline SARS-CoV-2–positive, and the median time from the last mRNA COVID-19 vaccination to study vaccination was 303 days (range, 154–675 days). All participants had received at least three previous mRNA COVID-19 vaccine doses; 92.0%, 30.1%, and 1.2% of participants had received four, five, or six previous doses, respectively. Medical history of participants at baseline is summarized in Appendix A. Demographic results for the variant neutralization subset are provided in Appendix A.

### 3.2. Safety

Local reactions reported within 7 days of the XBB.1.5-adapted BNT162b2 vaccination were mild to moderate in severity (Figure 2). The most common local reaction was injection site pain, which was less frequently reported among >55-year-old participants (52%) than in 12–17-year-old (80%) and 18–55-year-old (76%) participants. Median onset and duration of local reactions was from 1 to 2 days and from 1 to 3 days, respectively.

Within 7 days of vaccination with XBB.1.5-adapted BNT162b2, systemic events were predominantly mild to moderate in severity (Figure 3). Fatigue and headache were the most common systemic events, and both were reported less frequently among >55-year-old participants (35% and 26%, respectively) than among 12–17-year-old (57% and 37%) and 18–55-year-old (57% and 44%) participants. Fever was more frequently reported in 12–17-year-old participants (5/30 participants (17%)) than in >18-year-old participants (15/380 (4%)); however, due to the small number of 12–17-year-old participants included in the study, the point estimate lacks precision. Overall, two participants (0.5%) experienced fever from >38.9 °C to 40.0 °C (39.1 °C fever reported in the 12–17-year-old group and 39.0 °C fever in the >55-year-old group); the fevers occurred on Day 2 after vaccination and resolved after 1 and 2 days, respectively. No fevers >40.0 °C were reported. A further three participants reported severe systemic events, all in the 18–55-year-old group: two participants reported severe fatigue starting on Day 2, which resolved after 1 and 3 days, respectively, and one participant reported severe diarrhea starting Day 5 after vaccination. This event had been moderate from Day 2 after vaccination and returned to moderate severity on Day 6 but did not resolve until 20 days after onset. Median onset and duration of systemic events were 2 days and from 1 to 2 days, respectively.

Adverse events and serious adverse events reported within 1 month of the XBB.1.5-adapted BNT162b2 vaccination are summarized in Figure 4. Adverse events were infrequent (7.5% in the total population), and none led to study withdrawal. One participant >55 years of age with a history of paroxysmal nocturnal hemoglobinuria (PNH) who received five previous doses of mRNA COVID-19 vaccine reported serious adverse events of hyponatremia, acute kidney injury, and PNH after the XBB.1.5-adapted BNT162b2 vaccination; none of the events was considered related to study vaccination as determined by the investigator. At least one adverse event of special interest was reported by five participants (1.2%) through 1 month after receipt of the XBB.1.5-adapted BNT162b2; these included single reports of arthralgia, chest discomfort, and chest pain considered possibly vaccine related by the investigator; single reports of dyspnea and heavy menstrual bleeding considered possibly vaccine related by the investigator; and the single report of PNH as described above. No confirmed myocarditis or pericarditis reports and no cases of severe COVID-19 or of MIS-C were reported at the time of this analysis.

### 3.3. Immunogenicity

The evaluable immunogenicity population comprised 37 participants; three participants from the variant neutralization subset were excluded from this population because of protocol deviations on or before the 1-month postvaccination visit (18–55-year-old group, *n* = 1; >55-year-old group, *n* = 2).

The SARS-CoV-2 FFRNT 50% neutralizing titers against Omicron XBB.1.5, EG.5.1, and BA.2.86 were increased 7 days after vaccination with the XBB.1.5-adapted BNT162b2 compared with baseline levels, and were numerically higher than those in the matched comparator group of participants who had received the BA.4/BA.5-adapted BNT162b2 (Appendix A). The SARS-CoV-2 FFRNT 50% neutralizing titers against Omicron XBB.1.5, EG.5.1, and BA.2.86 were further increased 1 month after vaccination with the XBB.1.5-adapted BNT162b2 compared with baseline levels (GMTs and GMFRs are shown in Figure 5). GMTs at 1 month were similar for all three sublineages (GMT range overall, 452.5–561.3). Baseline GMTs were higher for Omicron BA.2.86 compared with XBB.1.5 and EG.5.1; GMFRs from baseline to 1 month after vaccination were similar for XBB.1.5 (overall GMFR, 7.0 (95% CI, 4.3, 11.4)) and EG.5.1 (8.7 (5.4, 14.1)) and slightly lower for BA.2.86 (4.5 (3.1, 6.5)). Postvaccination titers were generally higher in participants >55 years of age (GMT range, 559.3–732.3) compared with those from 18 to 55 years of age (370.3–444.4), while the baseline titers were generally similar in the two age groups.

Seven days after vaccination, the overall percentage of participants with seroresponse was numerically higher after the XBB.1.5-adapted BNT162b2 (overall range, 45.0–62.5%) than after the BA.4/BA.5-adapted BNT162b2 (17.5–25.0%; Appendix A). One month after vaccination with the XBB.1.5-adapted BNT162b2, the percentages of participants overall with a seroresponse to Omicron XBB.1.5 and EG.5.1 were similar (67.6% and 73.0%, respectively) and slightly higher than to BA.2.86 (59.5%; Figure 6). The percentages of participants with a seroresponse to Omicron XBB.1.5 were similar in 18–55-year-old and >55-year-old participants (68.4% and 66.7%, respectively). The percentages of participants with a seroresponse to Omicron EG.5.1 and BA.2.86 were slightly lower for 18–55-year-olds (68.4% and 52.6%, respectively) than for >55-year-olds (77.8% and 66.7%).

## 4. Discussion

In this analysis of 1-month safety and preliminary immunogenicity data from a substudy of an ongoing, open-label, phase 2/3 study, the XBB.1.5-adapted BNT162b2 in COVID-19 vaccine-experienced individuals ≥12 years of age had a safety and tolerability profile similar to that seen with original and the BA.5/BA.5-adapted and BA.1-adapted BNT162b2 vaccines [28,29,30,31] and induced substantial increases in neutralizing responses against Omicron XBB.1.5, EG.5.1, and BA.2.86.

The Omicron sublineages have predominated the SARS-CoV-2 epidemiologic landscape since November 2021, although variability in geographic distribution has been observed and attributed to social networking patterns, as well as the biological properties of the specific sublineage [1]. The SARS-CoV-2 Omicron XBB.1.5 sublineage has increased transmissibility compared with previous circulating sublineages, most likely because of immune escape and enhanced infectivity compared with previously circulating strains [1,32]. Other Omicron sublineages have more recently emerged, including EG.5 and its sublineages EG.5.1, EG.5.1.1, and EG.5.2 in February 2023; BA.2.86 in July 2023; and JN.1 in August 2023 [6,7,33,34,35]. Characterization of the biological properties of these more recent sublineages are forthcoming, including clarification on their transmissibility and virulence. Together, these observations emphasize the continued and rapid evolution of SARS-CoV-2 [36] and underscore the importance of continued surveillance and viral characterization as well as the availability of safe and effective vaccines to protect against the serious consequences of COVID-19.

The XBB.1.5-adapted BNT162b2 vaccine was safe and tolerable, with local reactions and systemic events within 7 days of vaccination of mild to moderate severity, adverse events that were infrequent and none of which led to study withdrawal, and no confirmed reports of myocarditis or pericarditis. The safety and tolerability profile of the XBB.1.5-adapted BNT162b2 is consistent with the profiles of original BNT162b2 and the BA.4/BA.5-adapted BNT162b2 [30,31]. In safety surveillance from the US Centers for Disease Control (CDC) using V-safe and the CDC and FDA Vaccine Adverse Event Reporting System (VAERS) from 22 September 2021, to 6 February 2022, in individuals 18 years and older who received more than 330,000 booster doses of original BNT162b2, the rates of local reactions and systemic events were consistent with those reported in our analysis of this current substudy of a clinical trial of the XBB.1.5-adapted BNT162b2 [30]. Additionally, in safety surveillance using the same databases in the first 7 weeks of the BA.4/BA.5-adapted BNT162b2 vaccine availability (i.e., from 31 August 2022 to 23 October 2022), at which time approximately 14.4 million individuals 12 years and older had received a BA.4/BA.5-adapted BNT162b2 booster dose, the frequency of local reactions and systemic events decreased with increasing age [31]; this is consistent with our analysis of the XBB.1.5-adapted BNT162b2. Longer-term follow-up in our trial and further uptake of the XBB.1.5-adapted BNT162b2 within global vaccination programs will provide further characterization of the safety and tolerability profile of the vaccine.

In September 2022, it was estimated that more than 96% of individuals ≥16 years of age in the United States had SARS-CoV-2 antibodies from previous infection or vaccination, including 23% from infection alone, 26% from vaccination alone, and 48% from hybrid immunity [37]. In spite of the epidemiologic setting of high population seropositivity, it is likely that vaccine-induced immunity will be reduced whenever antigenic drift causes a mismatch between the composition of COVID-19 vaccines and circulating SARS-CoV-2 strains [9]. Encouraging data in a subset of participants in this study indicate that the XBB.1.5-adapted BNT162b2 vaccine induces robust neutralizing responses to Omicron XBB.1.5 and EG.5.1 and slightly lower but still robust responses to BA.2.86. Seven days after vaccination, GMTs, GMFRs and seroresponses were numerically higher for all three sublineages following the XBB.1.5-adapted BNT162b2 than for the matched participants who had received BA.4/BA.5-adapted BNT162b2. Although the analysis was not powered to detect a difference between groups, the results are encouraging and in line with a preclinical study reporting higher neutralizing titers with the XBB.1.5-adapted BNT162b2 compared with the BA.4/BA.5-adapted BNT162b2 for XBB.1.5, EG.5.1, BA.2.86 and other sublineages [38]. One-month postvaccination titers against all three strains tended to be higher in participants >55 years of age compared with those from 18 to 55 years of age; the reasons for this have not been elucidated, and the sample size is relatively small. Immunogenicity differences between age groups will be further explored when data are available for all participants.

Limitations of this study include the short follow-up for both safety and immunogenicity through 1 month after vaccination. However, it is important to disseminate early and timely results to understand the potential benefit of the XBB.1.5-adapted BNT162b2, especially given that vaccination with the XBB.1.5-adapted vaccine is underway. Longer-term follow-up from this, including adverse events of special interest, and other studies is ongoing and will be supplemented with real-world effectiveness investigations. The immunogenicity analyses were conducted in a small subset of participants using an exploratory SARS-CoV-2 neutralization assay. Comprehensive immunogenicity results with a larger number of participants will be published later. Additionally, the study population was predominantly White, exclusively from the United States, and 12 years and older. Phase 3 trials of the XBB.1.5-adapted BNT162b2 are underway in children 5 to <12 years old and planned in infants and children 6 months to <5 years of age (ClinicalTrials.gov Identifier: NCT05543616). Previous trials in individuals from 6 months of age evaluating BNT162b2 variant-adapted vaccines have shown a similar safety and tolerability profile to the initial primary series with original BNT162b2 [39,40].

In conclusion, these safety and immunogenicity data support administration of the XBB.1.5-adapted BNT162b2 in vaccine-experienced individuals ≥12 years of age. Safe and effective variant-adapted COVID-19 vaccines closely matched to circulating strains will likely remain critical to protect the vulnerable against serious COVID-19 outcomes and to reduce the burden on healthcare systems.

## Figures and Tables

**Figure 1 vaccines-12-00118-f001:**
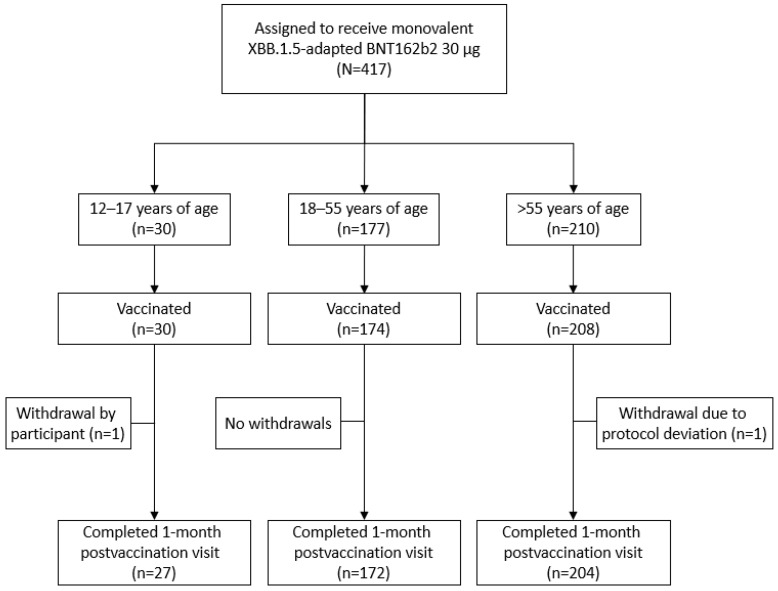
Participant disposition. At data cutoff (27 September 2023), some participants had not reached the 1-month postvaccination visit.

**Figure 2 vaccines-12-00118-f002:**
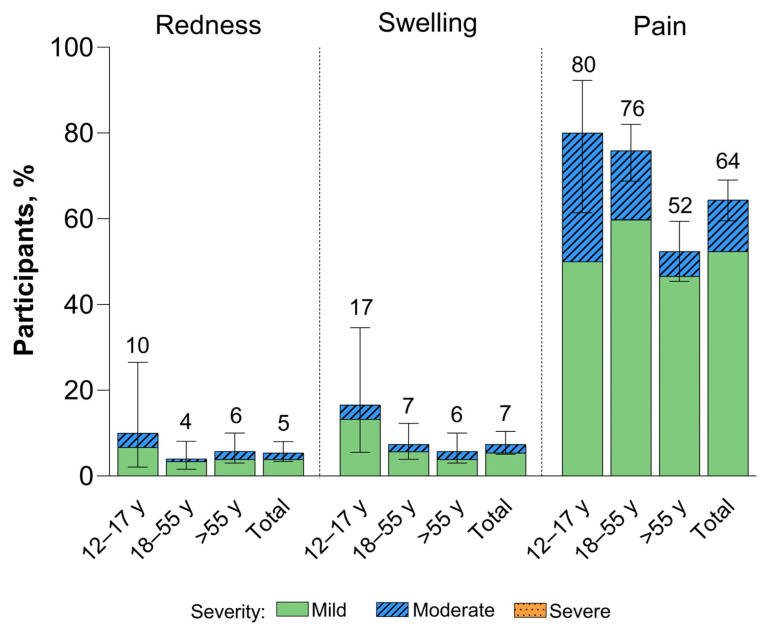
Local reactions occurring within 7 days after receipt of XBB.1.5-adapted BNT162b2 30 μg (safety population). Error bars and numbers above the error bars denote the 95% CIs and the percentage of participants in each group reporting the specified local reaction, respectively. The *N* values are 30, 174, 206, and 410 for 12–17-year-olds, 18–55-year-olds, >55-year-olds, and the total population, respectively.

**Figure 3 vaccines-12-00118-f003:**
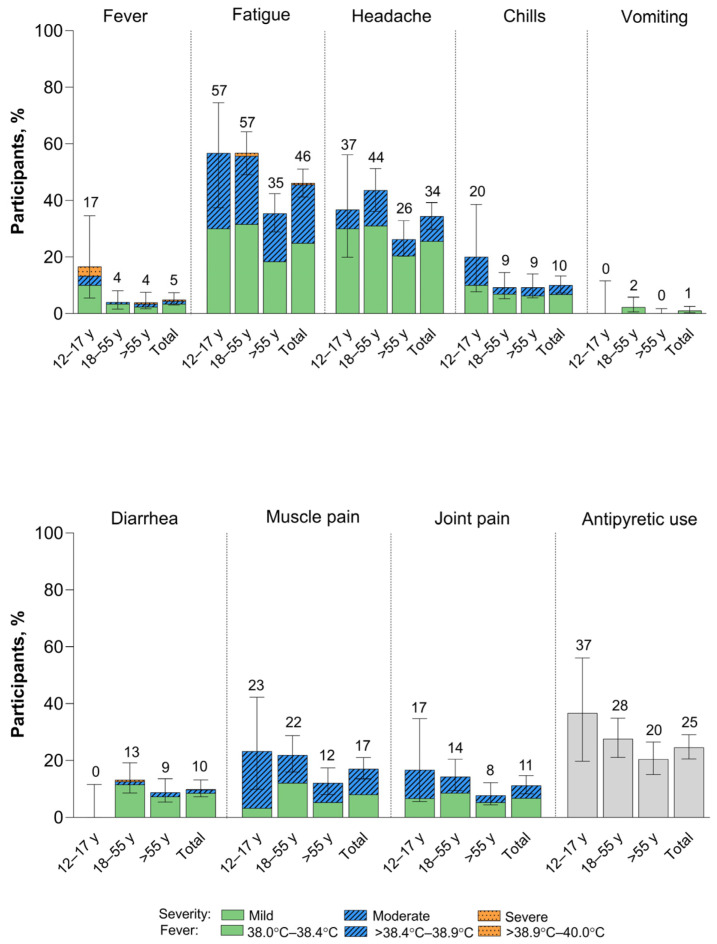
Systemic events occurring within 7 days after receipt of XBB.1.5-adapted BNT162b2 30 μg (safety population). Error bars and numbers above the error bars denote the 95% CIs and the percentage of participants in each group reporting the specified systemic event, respectively. The *N* values are 30, 174, 206, and 410 in 12–17-year-olds, 18–55-year-olds, >55-year-olds, and the total population, respectively.

**Figure 4 vaccines-12-00118-f004:**
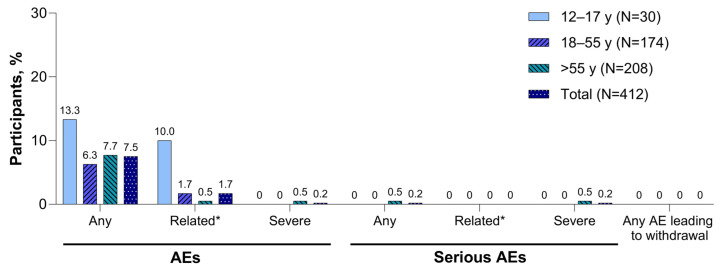
Adverse events reported within 1 month of XBB.1.5-adapted BNT162b2 30 μg (safety population). AE = adverse event; SAE = serious adverse event. * Related AEs and related serious AEs were as determined by the investigator at the clinical site.

**Figure 5 vaccines-12-00118-f005:**
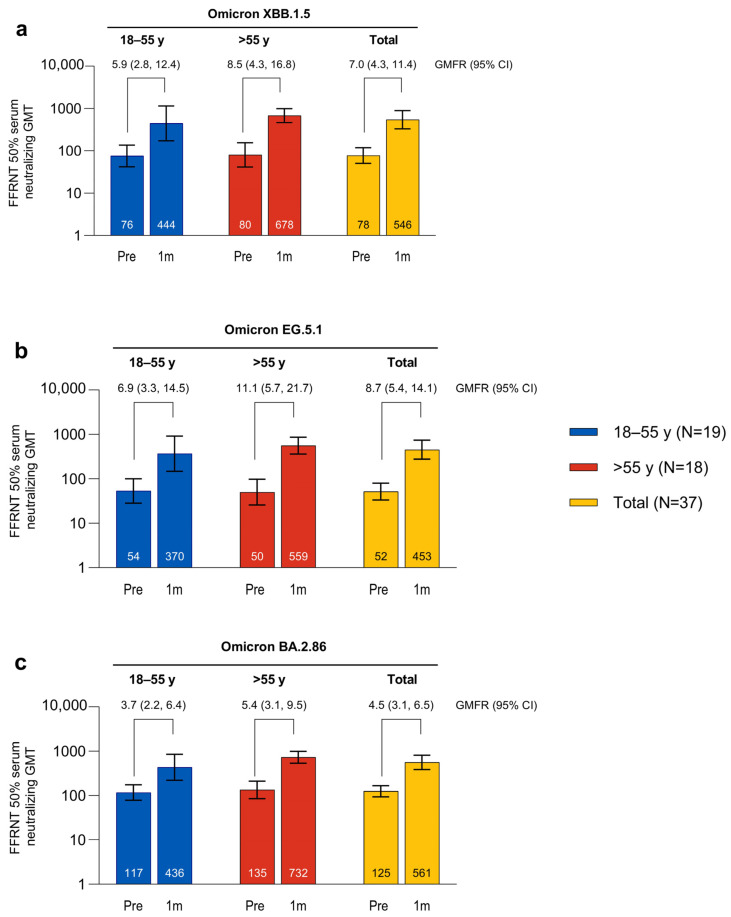
Serum-neutralizing GMTs (95% CIs) before and 1 month after vaccination with XBB.1.5-adapted BNT162b2 30 μg and GMFRs (95% CIs) from before to 1 month after vaccination to Omicron XBB.1.5 (**a**), EG.5.1 (**b**), and BA.2.86 (**c**). Data are for the evaluable immunogenicity population. Assay results <LLOQ were set to 0.5 × LLOQ. Numbers within the bars are the GMTs. 1m = 1 month after vaccination; FFRNT = fluorescent focus reduction neutralization test; GMFR = geometric mean fold rise; GMT = geometric mean titer; LLOQ = lower limit of quantitation; Pre = before vaccination.

**Figure 6 vaccines-12-00118-f006:**
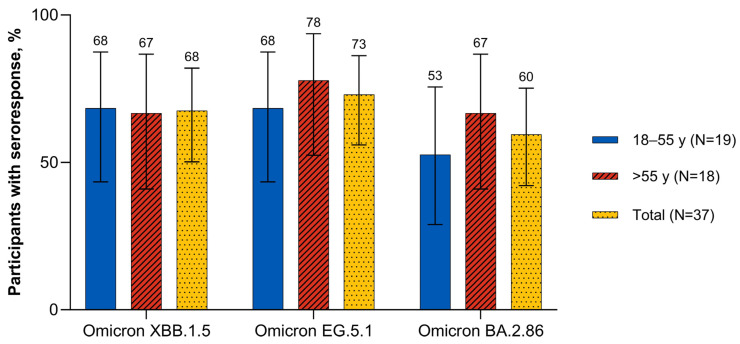
Participants achieving seroresponse (95% CIs) 1 month after vaccination with XBB.1.5-adapted BNT162b2 30 μg (evaluable immunogenicity population). A ≥4-fold rise from before study vaccination in FFRNT 50% serum neutralizing titers was considered a seroresponse. For participants with a baseline measurement <LLOQ, a postvaccination assay result of ≥4 × LLOQ was considered a seroresponse. FFRNT = fluorescent focus reduction neutralization test; LLOQ = lower limit of quantitation.

**Table 1 vaccines-12-00118-t001:** Participant demographic and baseline clinical characteristics.

Characteristic	12–17 Years Old (*N* ^a^ = 30)	18–55 Years Old (*N* ^a^ = 174)	>55 Years Old (*N* ^a^ = 208)	Total (*N* ^a^ = 412)
Sex, *n* ^b^ (%)				
Male	11 (36.7)	74 (42.5)	85 (40.9)	170 (41.3)
Female	19 (63.3)	100 (57.5)	123 (59.1)	242 (58.7)
Race, *n* ^b^ (%)				
White	26 (86.7)	135 (77.6)	165 (79.3)	326 (79.1)
Black	3 (10.0)	23 (13.2)	26 (12.5)	52 (12.6)
American Indian or Alaska Native	0	0	1 (0.5)	1 (0.2)
Asian	1 (3.3)	11 (6.3)	10 (4.8)	22 (5.3)
Native Hawaiian or other Pacific Islander	0	0	2 (1.0)	2 (0.5)
Multiracial or unknown	0	5 (2.9)	4 (1.9)	9 (2.1)
Ethnicity, *n* ^b^ (%)				
Hispanic/Latino	6 (20.0)	35 (20.1)	34 (16.3)	75 (18.2)
Age at vaccination, years				
Mean (SD)	14.0 (1.74)	40.4 (9.82)	68.6 (6.74)	52.7 (19.12)
Median (range)	14.0 (12–17)	42.0 (18–55)	68.5 (56–88)	56.0 (12–88)
Baseline SARS-CoV-2 status, *n* ^b^ (%)				
Positive ^c^	23 (76.7)	144 (82.8)	155 (74.5)	322 (78.2)
Negative ^d^	7 (23.3)	30 (17.2)	53 (25.5)	90 (21.8)
Prior COVID-19 vaccine doses, ^e^ *n* ^b^ (%)				
3 doses	30 (100.0)	174 (100.0)	208 (100.0)	412 (100.0)
4 doses	26 (86.7)	154 (88.5)	199 (95.7)	379 (92.0)
5 doses	1 (3.3)	19 (10.9)	104 (50.0)	124 (30.1)
6 doses	0	1 (0.6)	4 (1.9)	5 (1.2)
7 doses	0	0	1 (0.5)	1 (0.2)
Time from last dose of mRNA COVID-19 vaccine ^e^ to the study vaccination, months ^f^				
Mean (SD)	10.2 (1.95)	10.4 (2.33)	10.4 (1.82)	10.4 (2.05)
Median (range)	10.3 (6.7–12.6)	10.9 (5.5–24.1)	10.8 (5.8–22.6)	10.8 (5.5–24.1)
5 to <7, *n* ^b^ (%)	1 (3.3)	12 (6.9)	11 (5.3)	24 (5.8)
7 to <9, *n* ^b^ (%)	8 (26.7)	31 (17.8)	31 (14.9)	70 (17.0)
9 to 12, *n* ^b^ (%)	10 (33.3)	108 (62.1)	155 (74.5)	273 (66.3)
>12, *n* ^b^ (%)	11 (36.7)	23 (13.2)	11 (5.3)	45 (10.9)
Time from last dose of mRNA COVID-19 vaccine ^e^ to the study vaccination, days				
Mean (SD)	286.4 (54.50)	292.1 (65.24)	290.2 (50.95)	290.8 (57.54)
Median (range)	289.0 (188–352)	305.0 (154–675)	303.0 (162–634)	303.0 (154–675)
BMI, ≥16 years of age, *n* ^b^ (%)				
Number of participants ^g^	5	174	208	387
Underweight (<18.5 kg/m^2^)	0	4 (2.3)	2 (1.0)	6 (1.6)
Normal weight (≥18.5–24.9 kg/m^2^)	4 (80.0)	52 (29.9)	53 (25.5)	109 (28.2)
Overweight (≥25.0–29.9 kg/m^2^)	0	56 (32.2)	78 (37.5)	134 (34.6)
Obese (≥30.0 kg/m^2^)	1 (20.0)	62 (35.6)	75 (36.1)	138 (35.7)
BMI, 12–15 years of age/obese, ^h^ *n* ^b^ (%)				
Number of participants ^g^	25	-	-	25
Not obese	22 (88.0)	-	-	22 (88.0)
Obese	3 (12.0)	-	-	3 (12.0)

Data are for the safety population. BMI = body mass index; CDC = Centers for Disease Control and Prevention; NAAT = nucleic acid amplification test; N-binding = SARS-CoV-2 nucleoprotein–binding. ^a^
*N* was the number of participants in the specified group, or the total sample; this value was the denominator for the percentage calculations. ^b^
*n* was the number of participants with the specified characteristic. ^c^ Positive N-binding antibody result at baseline, positive NAAT result at baseline, or medical history of COVID-19. ^d^ Negative N-binding antibody result at baseline, negative NAAT result at baseline, and no medical history of COVID-19. ^e^ The inclusion criteria required the participant to have received ≥3 prior doses of a US-authorized mRNA COVID-19 vaccine, with the most recent dose being a US-authorized Omicron BA.4/BA.5–adapted vaccine ≥150 days before study vaccination. Four participants did not receive the US-authorized Omicron BA.4/BA.5–adapted vaccine; for these participants, the time from last dose was calculated from the last COVID-19 vaccine received before the study vaccination. ^f^ Month was calculated as 28 days. ^g^ For participants 12–15 years of age, obesity is defined as a BMI at or above the 95th percentile from the growth chart. Refer to the CDC growth charts at https://www.cdc.gov/growthcharts/html_charts/bmiagerev.htm (accessed on 18 December 2023). Because the participant age was expressed in years and the BMI values in the CDC growth charts are presented by age expressed in months, the minimum 95th percentile value for each age was chosen for the obesity criteria. ^h^ This value is the denominator for the percentage calculations for BMI/obese.

## Data Availability

Upon request, and subject to review, Pfizer will provide the data that support the findings of this study. Subject to certain criteria, conditions, and exceptions, Pfizer may also provide access to the related individual de-identified participant data. See https://www.pfizer.com/science/clinical-trials/trial-data-and-results (accessed on 18 December 2023) for more information.

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
