# Peer review of "Safety and Immunogenicity of the Monovalent Omicron XBB.1.5-Adapted BNT162b2 COVID-19 Vaccine in Individuals ≥12 Years Old: A Phase 2/3 Trial"

_vaccines, 2024, doi:10.3390/vaccines12020118_

Round 1

Reviewer 1 Report

Comments and Suggestions for Authors

This manuscript presents the early outcomes of the clinical trial phase II/III for the XBB.1.5 monovalent COVID-19 vaccine. Overall, the writing quality, technical aspects, methodology, and outcomes are outstanding and well-grounded

This manuscript is quite novel and will be very interesting in the field of COVID-19 vaccines.

Major concerns.

1. Comorbidities: It would be beneficial to clarify the comorbidities in Table 1 as certain underlying medical conditions might influence immunogenicity or vaccine efficacy/effectiveness, such as diabetes mellitus, obesity, and hypertension. 

If all recruited participants, except obese individuals, do not have any underlying medical conditions, it's recommended to provide clarification in the 2.1 Study Design and Participants section.

Minor concerns.

1. Consider creating an additional Figure to compare the FRNT 50% outcomes in each variant within specific age subgroups. This comparison would be valuable for readers to understand the reduction in GMT fold with other variants and to assess cross-reactions. 

The suggested figure should contain data for:
- 18-55 group: XBB.1.5, EG.5.1 and BA.2.86
- >55 group: XBB.1.5, EG.5.1 and BA.2.86
- Total group: XBB.1.5, EG.5.1 and BA.2.86

Comments.

1. Figures 2 and 3: While these figures appear to use a colorblindness-friendly palette, consider adding distinct patterns to each subgroup for better differentiation, particularly for readers who may print and read the figures in monochrome.

2. Figures 5 and 6: It is advisable to use a colorblindness-friendly palette or incorporate distinct patterns for better clarity. 

You can refer to the provided links for more information or select patterns that are distinguishable for colourblind readers.

http://www.cookbook-r.com/Graphs/Colors_(ggplot2)/#a-colorblind-friendly-palette

https://colorbrewer2.org/#type=sequential&scheme=BuGn&n=3

3. Line 85: Suggests using "(ClinicalTrials.gov Identifier: NCT05997290)" for clarity.

4. Line 147:  It would be beneficial to add clarification regarding the origin of the ID, such as "(ClinicalTrials.gov Identifier: NCT05472038)" for better understanding by readers.

5. Do some participants receive a viral vector or other vaccine platforms?

The heterologous or interchangeability may influence the immunologic outcomes. 
Various literature revealed that heterologous vaccination could increase the immunologic and efficacy/effective outcomes.

6. Suggest subgroup analysis by the Baseline SARS-CoV-2 status in each age group to make it more informative. You may be adding to the supplementary materials.

Participants with a history of infection may benefit from hybrid immunity, which could significantly enhance immunologic outcomes, including cross-reactivity with other variants.

Author Response

Please see attached Response to Reviewer 1

Reviewer 2 Report

Comments and Suggestions for Authors

In this paper, the authors report data on th esafety and immunogenicuty of the monovalent Omicron XBB.1.5-adapted BNT162b2 COVID-19 vaccine (called "MOAV" in the rest of this text. In total, 403 patients were included, belonging to different age groups. The authors conclude that MOAV has a saety and tolerability profile to that seen with the original and BA.5 & BA.1-adapted vaccines. 

Major comments:

1) The duration of follow-up of vaccinated people is not always very clear, in the material and methods it is mentioned a follow-up of 6 weeks for myocarditis and pericarditis, but apparently the follow-up was only carried out over a period one month; All this needs to be clarified. Furthermore, it is planned (p 3 line 130) that an evaluation be made within 6 months for the menstrual cycle), but there the observation times are too short. The material and method must therefore be reviewed accordingly to clearly separate what has been done and what must be done, the latter part being to be placed within the limitations of the study.

  2) The "materials and methods" section should be more precise, particularly concerning the methodology for selecting patients used for the comparative study BA.4/BA.5-adapted BNT162).

3) t is clearly indicated the number of anti-COVID vaccines received by the participants, but do the authors have information if the vaccines were from different laboratories or had always received the same type of vaccine. Indeed, some studies have shown that the immune response was greater when vaccines from different manufacturers were mixed. If this is not known, it should be added as a limitation to the study.

4) It says on page 9, line 247, "none of the events were considered related to study vaccination, as determined by the investigator"; It seems important to me, perhaps in the "additional data" section to describe in detail who the investigator is, how the evaluators worked (absence of link of interest with the manufacturer)

5) The summary should be revised, be more precise, by indicating precisely the duration of the study, the number of patients selected for the study, the aims of the study (which may explain why there is no no statistics on certain results since the study was not designed for this purpose)

Minor comment:

- fig 4, on the graph there is written "severe AE" and in the text it indicated SAE = seriuos adverse event"

Author Response

Author's Reply to the Review Report (Reviewer 2)

Top of Form

Review Report Form

Open Review

Quality of English Language

(x) I am not qualified to assess the quality of English in this paper
( ) English very difficult to understand/incomprehensible
( ) Extensive editing of English language required
( ) Moderate editing of English language required
( ) Minor editing of English language required
( ) English language fine. No issues detected

Yes

Can be improved

Must be improved

Not applicable

Does the introduction provide sufficient background and include all relevant references?

(x)

( )

( )

( )

Are all the cited references relevant to the research?

(x)

( )

( )

( )

Is the research design appropriate?

(x)

( )

( )

( )

Are the methods adequately described?

( )

(x)

( )

( )

Are the results clearly presented?

( )

(x)

( )

( )

Are the conclusions supported by the results?

( )

(x)

( )

( )

Comments and Suggestions for Authors

In this paper, the authors report data on th esafety and immunogenicuty of the monovalent Omicron XBB.1.5-adapted BNT162b2 COVID-19 vaccine (called "MOAV" in the rest of this text. In total, 403 patients were included, belonging to different age groups. The authors conclude that MOAV has a saety and tolerability profile to that seen with the original and BA.5 & BA.1-adapted vaccines. 

Major comments:

1) The duration of follow-up of vaccinated people is not always very clear, in the material and methods it is mentioned a follow-up of 6 weeks for myocarditis and pericarditis, but apparently the follow-up was only carried out over a period one month; All this needs to be clarified. Furthermore, it is planned (p 3 line 130) that an evaluation be made within 6 months for the menstrual cycle), but there the observation times are too short. The material and method must therefore be reviewed accordingly to clearly separate what has been done and what must be done, the latter part being to be placed within the limitations of the study.

Response: We have revised the methods and results and associated figures to clearly stipulate that safety follow-up was to 1 month. The limitations have also been revised to note that safety follow-up was to 1 month, and that longer-term follow-up from this study, including for adverse events of special interest, is needed.

  2) The "materials and methods" section should be more precise, particularly concerning the methodology for selecting patients used for the comparative study BA.4/BA.5-adapted BNT162).

Response: Participants from the comparative study were selected randomly and were matched by age, sex, and baseline SARS-CoV-2 status as summarized in Table S3 and Figure S1 and Figure S2.

3) t is clearly indicated the number of anti-COVID vaccines received by the participants, but do the authors have information if the vaccines were from different laboratories or had always received the same type of vaccine. Indeed, some studies have shown that the immune response was greater when vaccines from different manufacturers were mixed. If this is not known, it should be added as a limitation to the study.

Response: The study enrolled participants who had previously received at least 3 doses of an mRNA COVID-19 vaccine that was authorized by the US Food and Drug Administration. In addition to at least 3 mRNA COVID-19 vaccine doses, very few participants received other vaccine platforms and we have included this clarification in the Methods.

4) It says on page 9, line 247, "none of the events were considered related to study vaccination, as determined by the investigator"; It seems important to me, perhaps in the "additional data" section to describe in detail who the investigator is, how the evaluators worked (absence of link of interest with the manufacturer)

Response: Causality assessment for AEs and SAEs was the responsibility of the investigator at the clinical site. This clarification is included in the Figure 4 legend. In addition, if the investigator did not provide a causality assessment, then the event was to be labeled as “related to study intervention” for reporting purposes. However, there were no AEs for which an investigator’s assessment of relatedness was missing in the data reported in this manuscript.

5) The summary should be revised, be more precise, by indicating precisely the duration of the study, the number of patients selected for the study, the aims of the study (which may explain why there is no no statistics on certain results since the study was not designed for this purpose)

Response: We assume the reviewer is referring to the abstract in relation to the summary. All these items are currently stipulated in the abstract. We have also revised the opening paragraph of the discussion as follows: “In this analysis of 1-month safety and preliminary immunogenicity data from a substudy of an ongoing, open-label, phase 2/3 study, the XBB.1.5-adapted BNT162b2 in COVID-19 vaccine-experienced individuals ≥12 years of age had a safety and tolerability profile similar to that seen with original and the BA.5/BA.5-adapted and BA.1-adapted BNT162b2 vaccines [28-31] and induced substantial increases in neutralizing responses against Omicron XBB.1.5, EG.5.1, and BA.2.86.”

Minor comment:

- fig 4, on the graph there is written "severe AE" and in the text it indicated SAE = seriuos adverse event"

Response: This is correct as shown. There were severe adverse events (AEs) and severe serious adverse events (SAEs), which refers to the intensity of the event. Specifically, a severe AE or severe SAE was designated by the investigator as interfering significantly with the participant’s usual function, or significantly affecting clinical status, or requiring intensive therapeutic intervention.

Submission Date

19 December 2023

Date of this review

04 Jan 2024 14:29:05

Bottom of Form

Please see attached Response to Reviewer 2

Round 2

Reviewer 2 Report

Comments and Suggestions for Authors

Thank you to have answered to my comments.